# Insights into German Consumers’ Perceptions of Virtual Fencing in Grassland-Based Beef and Dairy Systems: Recommendations for Communication

**DOI:** 10.3390/ani10122267

**Published:** 2020-12-01

**Authors:** Ekaterina Stampa, Katrin Zander, Ulrich Hamm

**Affiliations:** Department of Agricultural and Food Marketing, Faculty of Organic Agricultural Sciences, University of Kassel, Steinstr. 19, 37213 Witzenhausen, Germany; k.zander@uni-kassel.de (K.Z.); hamm@uni-kassel.de (U.H.)

**Keywords:** agricultural innovation, animal husbandry, animal welfare, biodiversity communication, conservation marketing, consumer attitude, verbal protocol

## Abstract

**Simple Summary:**

Pasture-raised beef and dairy products are valued by consumers for their taste and higher animal welfare standards. Pasture grazing can be optimized using virtual fencing technology. The use of virtual fencing to guide cattle on pasture can contribute to biodiversity conservation by protecting environmentally sensitive areas. Concerns exist about consumers’ acceptance of virtual fencing in beef and dairy production. To explore consumers’ perception of virtual fencing, German consumers’ responses to information material about virtual fencing were analyzed. The results showed respondents’ uncertainty about the implementation of the technology with regard to its effects on animal welfare and possible social impact. Respondents showed readiness to support pasture grazing with their purchase decisions, yet struggled to see their personal advantages from the use of a specific grazing management practice. Thus, practitioners should consider keeping the focus in communication with consumers not on the technology but on tangible benefits, such as quality of pasture-raised products. Furthermore, state support is argued to be necessary to encourage livestock practitioners to adopt virtual fencing in cattle grazing for biodiversity conservation.

**Abstract:**

The share of cattle grazing on grassland is decreasing in many European countries. While the production costs of intensive stall-based beef and dairy systems are usually lower per kg product, grazing-based systems provide more ecosystem services that are valued by consumers. Innovative grazing systems that apply virtual fencing technology can improve animal welfare, optimize grassland use as pasture, and contribute to biodiversity conservation. Although consumer demand for pasture-raised products could promote animal-friendly practices, consumer perception of virtual fencing remains unknown. To address this gap in research, this study developed information brochures with different lines of argumentation and tested the responses of German consumers using concurrent think aloud protocols. The results demonstrated ambivalence in consumers’ attitudes to virtual fencing. The participants supported the idea of cattle pasturing to promote animal welfare and foster biodiversity declaring a willingness to contribute not only by paying price premiums for pasture-raised products but also through seeking other possibilities of action and participation. However, participants raised concerns about the effects on animal welfare and the social ramifications of the technology. The study offers recommendations for addressing these issues in communication and further contributes to the understanding of consumers’ perceptions of innovation in animal production.

## 1. Introduction

Further decline in pastureland and dairy cattle grazing in the EU is expected in the coming years [1,2]. It proceeds in spite of evidence that the traditional use of grasslands for grazing contributes significantly to improved animal welfare [3], biodiversity and cultural landscape conservation [4], and the attractiveness of life in rural areas [5,6]. To counteract the decrease in the share of pastureland in total grassland, it will be important to ensure governmental support in motivating farmers to adopt sustainable grazing practices and compensating them for the higher production costs incurred in adhering to higher animal welfare standards and conserving biodiversity [7]. While a range of governmental policy instruments can be applied, such as taxes and subsidies, an alternative mechanism involves creating markets by providing information to consumers [8]. Given that increasing consumer demand for pasture-raised beef and dairy products could contribute to the sustainable use of grasslands and the conservation of their ecosystem services [1], it is important to explore consumers’ perceptions of grazing-based livestock production.

For grazing to be economically viable, it requires active management. Compared to continuous grazing, rotational grazing management systems that involve splitting pasture into paddocks allow more effective use of fodder and can enhance insect biodiversity [9]. One of the developing technologies aimed at optimizing grazing management is virtual fencing, which is an automated instrument for cattle herding based on restricting movement through audio warning tones and electric signals in the absence of tangible boundaries [10]. Virtual fencing can be potentially used for cattle management in the areas where traditional fencing is not possible and for conservation of environmentally sensitive areas within pastures, to protect the nests of endangered birds and rare plant species located there [11,12]. At the same time, virtual fencing can reduce costs related to the installation, maintenance and relocation of electric fences [13], thereby enabling farmers to expand pasture grazing for their cattle or to adopt it in the first place. Current solutions employ a combination of GPS technology to track animals in the landscape with an electronic neckband worn by cattle that emits warning signals. Research in the field of virtual fencing has so far been limited to technical aspects, animal ethology and welfare (e.g., References [13,14]). Consumers’ perspectives on virtual fencing are as yet unclear, though concerns exist as to its socio-ethical acceptability [15]. This finding is supported, for example, by the results of a study of Dutch citizens’ views on modern farming that indicates a preference for a rather traditional, natural dairy farming [16]. Whereas the attitudes of consumers to agricultural innovations have been explored in relation to more established fields, such as genetically modified organisms and automatic milking systems (e.g., References [17,18]), the road to understanding consumers’ perceptions of virtual fencing in pasture grazing systems has yet to be paved.

Research on agricultural innovation suggests that the use of virtual fences could be a contentious socio-ethical issue, on the one hand, perceived as beneficial for animal welfare and biodiversity but also seen as being ‘unnatural’ and overly technical [15]. Contrary to these latter concerns, however, it has been shown that the welfare and behavior of cattle in grazing systems that use virtual fencing is no different than in systems that use electric fencing [10], while pasture grazing itself positively affects cattle welfare [3]. These facts are not known by most consumers, however, as there is a widespread lack of public knowledge about agricultural practices and animal welfare [19,20]. In the absence of such knowledge, consumers have been found to weigh up the acceptable and inacceptable aspects of a technology rather than clearly stating an attitude [17]. Efforts to address this by providing excessive information on the new technology and on conservation may prove counterproductive, however, actually serving to increase consumer uncertainty and confusion [21,22]. Nevertheless, many consumers do show an interest in agricultural topics and actively seek and obtain information on these topics from mass media [20,23], which in its turn can shape consumers’ attitudes and, thereby, affect their behavior [24].

Addressing consumers’ values underlying their behavior is crucial for successful marketing of products from an innovative grazing system. Whereas the advantages of the new technology and biodiversity conservation may seem somewhat abstract to many consumers, pasture-based systems also provide tangible benefits in form of high-value food products. As a recent review has shown, consumers appreciate the improved animal welfare and lower environmental impact of pasture-raised products [25]. They associate improvements in animal welfare with better product quality, taste and healthiness [26]. This motivates substantial consumer segments to be willing to pay higher prices for meat and dairy from ethical production systems with lower environmental impact and better animal welfare [27,28]. Furthermore, landscape attributes, such as the presence of grazing animals, provided by grasslands used for grazing are likely to be appreciated by consumers [29]. German consumers also support measures aimed at animal welfare improvement, the creation of innovative forms of financing for nature conservation, and an expansion of nature protection areas [30,31]. For many consumers, however, it may prove difficult to change their consumption habits and behavior, despite their understanding the competitive disadvantages arising for farmers from the adoption of pasture grazing, stricter animal welfare standards, and biodiversity conservation measures [32,33]. Consumers’ behavior regarding pasture-raised products from the new grazing system is likely to be affected by factors, such as sense of own self-efficacy, social norms, feelings, and cognition, as well as perceived personal benefits, personal relation to food production, the associated costs of the products, and their availability [34,35,36,37,38,39]. Addressing at least some of these factors using effective targeted communication and appealing visual and textual materials can increase the inclination of consumers to consciously perceive and process information on pasture-raised products and their propensity toward pro-environmental behavior [36,37,38,39]. Seeing that European consumers agree on the importance of better communication and information provision on the issues of animal welfare and biodiversity [31,40], there is considerable potential for livestock producers to market pasture-raised products for higher prices and thereby gain the support of consumers for sustainable agricultural innovations.

The influence of different communication approaches and textual and visual information have been previously studied with regard to consumers’ perceptions of animal husbandry and welfare (e.g., References [23,41,42,43]). However, considering consumers’ possible ethical concerns regarding animal welfare in virtual fencing systems, it is of interest to explore the communication potential of its aspects related to nature conservation. While a few recent studies have addressed different aspects of communication of biodiversity conservation [44,45,46,47,48], no studies (to the authors’ knowledge) have yet explored communication of biodiversity conservation through the lens of beef and dairy production, especially in the context of innovative practices in animal husbandry. Given the identified research gaps, insights from a closer exploration of consumers’ perception of the innovative grazing system can be useful to inform practitioners’ decisions when developing communication strategies. The aim of this study was thus to explore consumers’ perceptions, understanding, and acceptance of virtual fencing in pasture-based livestock production, focusing on aspects related to the technology, animal welfare, pasture-raised products, and environment and biodiversity. The research objectives and questions are presented in Figure 1.

## 2. Materials and Methods

The postpositivist research paradigm was adopted in the preparation, conduct and reporting of this study. Figure 2 depicts the study design. The study is reported in adherence to the standards for reporting qualitative research [49].

### 2.1. Test Object

Since flyers and folded brochures serve as an information source for large consumer segments prior to their purchase of meat products in German supermarkets [42,50], information brochures were used to stimulate consumers’ reflection on the novel concept. Four tri-fold brochures were crafted, all entitled (in German) ‘Pasture Cattle Farming’. Each brochure had a separate topic announced on the cover and a different cover image to illustrate the topics: ‘Good for animals and humans’, ‘Promoting animal welfare’, ‘Conserving biological variety’, and ‘Conserving natural landscapes’.

Drawing on the recent findings in conservation communication and the work of Schaffner et al. [48], several communication approaches were used. A cognitive-informational approach was realized through both declarative knowledge (facts, trends, arguments) and procedural knowledge (possibilities of action, suggestions, advice) strategies. In terms of the emotional-experiential approach, positive emotionalization (positive emotion strategy) was attempted using visual stimuli, including appealing motifs of nature, animals, and people, and a bright pallet of blue, green, and white. An effort was made to provide balanced or neutral textual information. The negative emotion strategy was omitted. In terms of benefit strategy, each brochure included suggestions of the personal benefits of biodiversity conservation and animal welfare. Two of the brochures also highlighted the individual contribution of consumers. The moral norm strategy appealed to consumers’ responsibility for nature and for future generations.

Varying levels of argumentation about animal welfare, sensory enjoyment, biodiversity, landscape, farmers, product quality, and virtual fencing were selected to evoke a wider array of associations and to reduce the impact of specific designs on participants’ responses. The term ‘biodiversity’, previously found unsuitable in German language for communication [51], was only mentioned in contact information, while ‘biological variety’ was used throughout the texts.

### 2.2. Data Collection

#### 2.2.1. Think Aloud Protocols

To provide insights on consumer perception, explorative data were collected using concurrent think-aloud protocols (TAPs). This method draws on the verbalization of participants’ thoughts during the performance of tasks, such as reading texts in order to provide insights into thought processes as they are happening [52,53]. Compared to structured interviews, TAPs allow participants to explore the test object at their own pace and to go beyond any assumption on the part of the researcher [54]. Regarding the test object in this study, freedom to explore is essential because it allows to observe which visual or textual elements provoke consumers’ response. This method also provides fuller data than retrospective TAPs [52,54] and helps to avoid false recall [53]. The limitations of the method of concurrent TAPs are susceptibility to lack of realism, lack of generalizability and possible data incompleteness due to cognitive overload and small sample sizes of 10 to 30 [53,54]. TAPs have been used in consumer studies, inter alia, to explore consumer awareness of sustainable aquaculture products [55] and motivation to engage with the labeling of food products [56]. On the basis of the observed benefits and the versatility of this method in different study designs, concurrent TAPs were used in this study to collect explorative data on participants’ perceptions of information.

#### 2.2.2. Test Procedure

A two-phase pre-test was conducted to assess the comprehensibility and design of the brochures and to test the interview guide. After final revision, the brochures were typeset and printed by an external professional. The TAPs were recorded in November–December 2019 in three German cities: Osnabrück (North-West), Cottbus (East), and Augsburg (South). These cities were selected on the basis of their average purchasing power index [57], their population, and their location in areas of widespread pasture grazing. The study participants were recruited from pedestrian streets in the city centers. Purposive quota sampling was applied, with equal quotas set for participants aged 18–49 years and for participants aged 50 years and older in accordance with the median age of the German population at between 18 and 80 years of age [58]. The quota of female participants was set to two-thirds on the basis that women in Germany are more often responsible for the purchase of food in private households [59].

In the first stage of recruitment, a trained recruiter systematically approached every third passer-by. The screening criteria included regular consumption of beef and dairy products and no personal affiliation to agriculture or the food industry. The participants who fulfilled these criteria were invited to an interview and informed about the study background, the interview recording, the anonymity of data handling, and the amount of monetary compensation for their time. As the data collection progressed, the recruiter only approached passers-by to fill the missing quota.

All the TAPs were conducted by the first author, a female researcher trained in qualitative data-collection and analysis. The only persons present during the TAPs were this researcher and a single participant at a time. Once the participants had given their oral consent, the researcher switched on a voice-recorder and a camera focused on the brochures. The study began with a training phase to acquaint participants with the method; after that, a task was given to examine the four brochures, presented in randomized order. Participants were asked to verbalize every thought that came to their mind, to read aloud each text they were currently reading, and to vocally refer to the images they were looking at. Interactions between the researcher and the participants during the TAPs were minimized and aimed solely at encouraging continuous verbalization. Subsequently, a brief interview was conducted to elicit any thoughts the participants wished to add to their verbalizations, their understanding of virtual fencing, and their assessment of the TAP method. A pen-and-paper questionnaire was used to collect demographic data and the participants’ experience of pasture-raised products.

All research steps were thoroughly documented and the audio and video recordings of the TAPs were supplemented with the interviewer’s fieldnotes from each TAP session and the data from the brief interviews conducted after each TAP. The fieldnotes focused on the participants’ observable behavior and were used in both the protocol transcriptions and in the composition of case summaries.

#### 2.2.3. Sample Description

The total sample of 20 participants comprised twelve women and eight men, equally distributed between age groups, with the age of the participants ranged between 22 and 66 years old (see Table 1). The share of female participants deviated slightly from the initial quota. Six persons were recruited in the East, and seven participants each in the North-West and the South. Most participants in this study were university graduates.

Most of the participants had purchased pasture-raised products in the past or had tried them on occasions of out-of-home consumption. Some participants were uncertain about their previous consumption of pasture-raised products, while only a few were certain of never having purchased or tried these products.

### 2.3. Data Analysis

All data were transcribed verbatim by two trained assistants according to guidelines based on Kuckartz [60]. The transcripts were managed using MAXQDA 11 software (VERBI GmbH, Berlin, Germany). Thematic qualitative text analysis was employed to analyze the transcripts, a method based on structuring and summarizing data using a coding frame consisting of concept- and data-driven categories [60]. Thematic analysis is applied for problem-centered interviews and focus groups, making it suitable for verbal protocols.

An iterative process was used to build the coding frame. Concept-driven categories were extracted from the brochures, while data-driven categories and subcategories emerged from the fieldnotes and from the initial reading of the first transcripts. The categories were defined in a coding manual to ensure unambiguous coding. The smallest coding segment was a sentence. As data collection and transcription progressed, more data-driven categories emerged. In the next stage the first author and a trained assistant independently coded five transcripts, adhering to consensual coding, and subsequently revised the coding frame. The intercoder agreement coefficient, calculated using a MAXQDA built-in tool, exceeded 82%, thus confirming the reliability of the coding frame [61]. The first author then applied the coding frame to the remaining transcripts.

## 3. Results

The idea of pasture grazing and biodiversity conservation was welcomed by the majority of participants. Virtual fencing, however, raised a number of technical questions, critical remarks and concerns. Biodiversity conservation and animal welfare were often reflected on in relation to the new technology, and the social and individual aspects of the technology were also discussed, while explicitly agricultural topics received comparatively little attention (see Figure A1 for the coding frame). Excerpts from the transcripts are translated into English in this paper to illustrate the categories with typical statements. The texts that were read aloud from the brochures are in {curly brackets} and the source (in round brackets) provides the participant’s number, gender, their age group, and the passage number in the transcript.

### 3.1. Information Perception

Positive emotionalization through visual elements was actively noticed and appreciated by several participants. On the other hand, a few participants referred to the positive presentation of cattle farming as “wishful thinking”, even though they hoped it could become reality. The visual appeal of the cover images was in many cases decisive in participants’ choice of which brochures to read first, with brochures with “the least stimulating” covers examined last. Images of animals and landscapes were found appealing, though even positive depictions of grazing cattle, despite being pleasing, sometimes led to negative thoughts about intensive cattle farming. Images of people, insects, meat, and milk were often disliked and had no positive emotionalization effect (see Table 2). Familiarity with the depicted objects and situations had a positive effect on the extent to which participants liked each image:
“I think I like these pictures here most because they’re more familiar to me. Down there, this picture of a meadow, somehow, I find it very pretty. It’s…it’s more my thing.”(18f_50+; 18)

Two participants declared they would use the available QR code (Quick Response code) to access additional information while two other participants said they were unlikely to encounter brochures as a source of information in everyday situations. Thinking about their usual methods of information-seeking, younger participants named the Internet as their preferred information source, while most of the elder participants said they relied on their own experiences, personal communications, and television. Several participants perceived the information provided in the brochures as “food for thought” (5m_50+; 38) and a tool with which to reach out to persons presently unconcerned about their food choices. A few other participants skeptically referred to the brochures as advertising material for virtual fencing and/or pasture-raised products, yet found the information source credible:
“Well, you can always present/ make a great presentation of so many things in such brochures, and in the end it’s nothing more than window dressing […] Where [the funding body] is given then it’s/ then it’s scientifically supervised, I find it very good.”(1f_50+; 16)

### 3.2. Information Understanding

#### 3.2.1. Understanding of the Virtual Fencing Concept

Most participants understood the principle of virtual fencing and were able to describe it rather correctly when explicitly asked about their understanding of the concept, though some did admit a lack of comprehension, with one saying “I don’t understand it, but it’s also not really important to me” (18f_50+; 44). The graphic depiction of the virtual fencing principle was found helpful even in cases when a participant’s first reaction to it was confusion. The retainment of electric fences along the boundaries of pastures, which is an obligation under current agricultural insurance policies in Germany, was unclear to most participants, leading some to express doubts about the usefulness of virtual fencing. After some reflection, however, several participants did eventually understand the reasons for the current need to combine virtual fences with physical fences.

#### 3.2.2. Associations with the New Pasture Grazing System

All of the participants pondered the possible implications of virtual fencing in grazing systems. Their technical questions referred to the precision of cattle location measurement, the volume of the audio signal, cybersecurity, battery life, loading and changing, GPS and neckband failure management, and the technology’s effectiveness to control cattle stampedes. As some of the emerging questions were not immediately answered in the brochures, several participants voiced distrusting attitudes:
“And when the cattle get this [signal] and I’m standing right beside them, do I get the electric impulse too or what? I don’t really find it so trustworthy. […] My question is—doesn’t it do something to humans, and animals, and so on, in the long run?”(4f_50+; 11)

The connection between virtual fencing and biodiversity confused some: “What does it actually have to do with the biological stuff when everything is being done technologically anyway? I don‘t get it” (14m_< 50; 17). The presence of cattle in the landscape was welcomed as “a nice, pretty picture for people, too, to see the animals grazing in harmony with nature” (8m_50+; 18). The idea of recreation in nature, however, was regarded by some as “counterproductive” to the goal of biodiversity conservation. Some participants favored expanding protection measures in agriculture beyond animal welfare:
“{Conserving natural landscapes}, I would absolutely want that, that’s also my concern. This sounds good, but {Promoting animal welfare}—that wouldn’t be enough for me.”(1f_50+; 12)

Participants’ levels of interest in the subject of animal welfare in general were motivated by their dietary habits: “because we eat a lot of beef at home and it would be interesting to know how cows are doing” (2f_<50; 11). The reactions to the facts on the decline of pasture grazing provided in the brochures as part of the declarative knowledge strategy ranged from skepticism to shock. The association of animal welfare with the new grazing system was not obvious to the participants, however, with one commenting that: “If I were to think about animal welfare, I’d probably think of something else, something other than GPS neckbands” (7f_<50; 12). On the other hand, pasture grazing as such was referred to as a “natural”, “normal”, “right”, and “true” form of cattle husbandry, “like it used to be” and “rather rare”. It was associated with freedom of movement for animals by nearly half the participants, who saw it as a necessity for other animals, as well and appealed to moral norms:
“It doesn’t matter if it’s about dairy animals or laying hens […] space is such a thing for animals that shouldn’t be a luxury.”(19m_<50; 35)

Although no production systems other than pasture grazing were mentioned, participants often positively distinguished it from intensive animal farming. Pasture grazing was further seen as a solution to food-feed competition, for example, “because it’s nonsense to cut down the rainforest to plant soy beans there and then ship them over here” (4f_50+; 14).

### 3.3. Information Assessment

#### 3.3.1. Readiness to Accept Virtual Fencing in Beef and Dairy Systems

In many cases, instead of thinking about the acceptability of virtual fencing in grazing systems, participants attempted an estimation of its feasibility in Germany and expressed their doubts: “My first thought is that I believe it won’t be realized anyway” (7f_<50; 16). The reasons cited for such doubts included a lack of grasslands and available pastures near farms, general consumption patterns, bureaucratic issues, inadequate infrastructure, and low levels of public support. Feasibility was further connected to the implementation costs of the new technology, and, in this regard, the participants often referred to the need for governmental support, especially for small farms.

Participants perceived farmers’ work as “very, very hard work” (5 m_50+; 18) and expressed sympathy with farmers for the everyday hurdles and bureaucratic burdens they face. Accordingly, they were concerned about the implications of the novel grazing system in terms of increased unemployment, governmental surveillance and farmers’ workload:
“I really don’t think that a farmer can actually earn enough money in this way to secure a livelihood. I think if pasture fences are not used anymore and the cattle graze freely and there’s practically nothing to do, then the farmer’s profession will go extinct or become a very rare profession […] What happens to the farmers who lose their jobs in this way?”(5m_50+; 18, 38)

One participant noted that farmers contributing to biodiversity conservation by using new technology might have a positive effect on current discourse: “Perhaps it’ll also help put an end to this silly debate about farmers and agriculture being against nature protection” (18f_50+; 19).

The trend of digitalization and the use of large agricultural machinery were perceived rather negatively by participants in terms of their effects on the environment, though some opined that “[a] modern farm cannot do without technology” (6m_50+; 11). The difference between virtual fencing and the image of machine-intensive agricultural production was noted positively, though some participants said that the remote control implied by virtual fencing was “far from real life” and did not fit their view of farming. A tone of submission to digital technologies was also noticeable, however, with one younger male commenting on an image of a farmer with a tablet thus: “I have a cell-smart stuff myself […] my god, what else can one do? Everyone has to go along with it, somehow” (14m_<50; 34). Virtual fences were also perceived as sustainable since they “can be reused again and again” (10f_50+; 13).

Participants’ concerns and uncertainties that lowered the acceptability of virtual fencing included references to total surveillance and governmental control and the effects on human health of hormones consumed in meat from cattle subjected to psychological stress, as well as presumed effects on humans and insects of transmitted electronic signals and irradiation they associated with the technology. It was mostly elder participants who expressed sorrow about life being “too technical” and who voiced doubts as to the necessity of virtual fencing given the ubiquity of traditional fences. Arguments against virtual fencing included adherence to the status quo, e.g., “it has been this way for centuries” (10f_50+; 11), as well as concerns about employment in rural areas: “How about we employ a couple of cowherds again?” (20f_50+; 24). Safety concerns about virtual fencing referred to the safety of cattle from wild animals and trespassers, as well as to the safety of hikers being potentially endangered by free-roaming cattle, with one participant commenting that “a cow is not quite harmless” (12f_<50; 13). On the other hand, electric fences were described as possibly being unsafe for children and wild animals. Like safety, the aesthetic impacts of the system were brought up indirectly in participants’ arguments both for and against virtual fencing, with some preferring traditional fencing as being more “natural”, e.g., “When I build a fence in a natural way like it used to be done, with wood and stuff, then it’s not really that annoying in the landscape, in my opinion” (4f_50+; 11), and others favoring the new system over the unnaturalness of electric fencing, e.g., “Well, I find electric fences not so pretty, anyway. […] Such things have nothing to do in nature and I think that [virtual fencing] is a good alternative” (11f_<50; 11).

Many participants raised issues of animal welfare with regard to electric impulses, though only a few were concerned about audio warnings. Participants compared electric stimuli with shocks from electric fences, seeing them as “really not so bad” (16f_<50; 13), while some were ready to condone the use of electric impulses as long as they “are not so hard on the animals and don’t harm them” (14m_<50; 28), but also suggested a need for “caution” regarding different pain thresholds and the frequency of the signals. Several participants were unconvinced by the weak signal argumentation, “that’s basically the same [as electric fencing]” (7f_<50; 14), and appealed to naturalness and non-violence in animal control: “Well, I truly don’t know what it has to do with animal welfare when one frightens them [the cattle]. No matter by which means” (13f_50+; 11). For other participants, biodiversity conservation clearly took priority in the discussion of setting boundaries for animals:
“Sure, [cattle] must be enclosed somewhere, otherwise they’ll go everywhere. […] So, it’s part of ensuring a variety of species to/ Yeah, to set a limit. Definitely.”(16f_<50; 36)

Although the participants associated pasture grazing with better animal health, concerns were raised about the effects of virtual fencing on the health of cattle, especially regarding animals’ mental wellbeing. Most participants did not doubt the cognitive abilities of cattle to learn the association between a neckband cue and a boundary, but they emphasized the importance of a “gentle” training phase. Furthermore, animal welfare and environmental benefits were sometimes seen as competing goals:
“Well I do understand that a compromise between cattle welfare and environmental welfare is necessary. But I don’t know whether it still harms the one or the other. I don’t know that.”(3m_<50; 13)

#### 3.3.2. Readiness to Support Beef and Dairy Systems that Use Virtual Fencing 

Thinking aloud about their purchase decisions for beef and dairy products, a few participants confessed that factors, such as biodiversity and a sense of personal responsibility for a better future (as implied by the normative communication approach), were of no importance in their decisions due to the lack of time. Price premiums also remained a barrier for those who struggled first of all to provide enough food before they could consider purchasing sustainably produced beef or dairy products. Whereas some were concerned about generally rising living costs that left “not so much money for food” (2f_<50; 11), others expressed the view that “we spend too little on food” (18f_50+; 17). Higher prices were regarded by some as imperative to ensure high levels of animal welfare, though none of the participants connected price with biodiversity conservation measures in virtual fencing systems in their utterances. Many participants also noted that price would not be a major barrier if a high-quality product was consumed in modest amounts and with pleasure. They preferred quality over quantity and declared their willingness to pay higher prices to do something good for their own bodies by obtaining products of trustworthy origin that taste good and are produced under stricter animal welfare conditions. The latter two attributes were named as benefits for consumers by just over a half of all participants. Whereas a quarter of participants positively associated human health with cattle health, the benefits of ecosystem health supported through application of virtual fencing did not resonate with the participants. In general, argumentation for the new system based on personal benefits was mostly found insufficient:
“What benefits does it have for myself? Or are they only for cattle? Does it have benefits only for cattle or also for me as a consumer?”(5m_50+; 15)

With regard to beef and dairy products’ attributes, that may influence the readiness to support grazing systems that use virtual fencing, every single attribute was brought up and discussed by a variable number of the participants, but never by all of them, thus reflecting personal differences in the attribute importance. Over half of the participants reflected on the taste of pasture-raised products and agreed they “taste better”, positively associating the grazing of pasture grasses with “different” taste and “better meat”. While some participants based their judgement on hedonic experiences, others reported altruistic factors as affecting their enjoyment of the product “because you simply know that the cattle are well nurtured and have a good life” (15f_<50; 11). However, some doubted they could taste the difference between pasture-fed and corn-fed meat. When thinking about product quality, participants referred to the higher quality of pasture-raised products as an expected benefit. Expectations differed, however, for meat and dairy products. For instance, one participant who appreciated high-quality meat was not as demanding about milk: “Milk is milk. I don’t’ know/ It’s all from the same cow anyway, isn’t it?” (14m_<50; 17).

Participants reported a sense of loyalty to their usual shopping locations. Product availability in convenient shopping venues was important to them in choosing between more or less ethical products. When time pressure was absent, however, participants declared they were ready to literally go the extra mile for more sustainable options and expressed an interest in products from the new grazing system:
“The question is where can you get this beef and this milk from? […] This would be interesting [to know] where, where do you purchase […] the meat from cattle and [milk] from cows kept this way?”(16f_<50; 13)

The participants’ interest in products from the new grazing system included the query: “But how do I find them in a supermarket? How are they labeled?” (5m_50+; 16). Several other participants, mostly those who habitually shopped at farmers’ markets or at a butcher’s, reported having personally communicated with salespersons to get additional information that had increased their trust in the origin of the products they purchased. One such participant remarked that, even when a salesperson is trustworthy, there is still a need for certification and a traceability tool to ensure the transparency of the product’s journey from pasture to table:
“The problem that most people have, including myself, is that I don’t know and not even the sales lady at the butcher’s knows where the meat comes from. […] There must be a law, where/ so, this is controlled, the whole process, where the cow had grazed, where it was slaughtered.”(12f_<50; 12)

Thinking aloud about their individual contributions to environmental protection, participants mentioned their conscious food choices, with some expressing the view that “as a consumer, it is in fact important to support farmers” (6m_50+; 13). Participants were dissatisfied with the options for action suggested in the brochures as part of the procedural knowledge strategy and wished for more ways to contribute “other than that I really try to buy pasture-raised beef” (7f_<50; 14). Additional options for action brought up by participants included “spreading the word” about pasture-raised products and crowdfunding cattle for personal consumption. Concern about self-efficacy was present in some statements: for example, “I couldn’t change anything just by myself” (5m_50+; 17). Nevertheless, many participants agreed that everyone can contribute to a change in production conditions and “everyone should begin with oneself, and me with myself” (20f_50+; 33), since “when there’s demand there are people who will try to satisfy this demand, and then it happens so as it is now” (19m_<50; 39). Readiness to change one’s own dietary habits was seen by some participants as an initial contribution to environmental protection. Others suggested that biodiversity conservation should begin in one’s own backyard:
“Yes, this should be not only on a pasture but in private gardens, too. [That is] my opinion. So, this English-type lawn without any flower diversity/ It must begin with private households.”(8m_50+; 11)

A few participants talked of the need for reduced consumption in relation to their ideas about the consumption of meat by other citizens and criticized excessive consumption. Individual contribution was also frequently associated with reduced meat and dairy consumption in favor of ethically produced quality products:
“Then you drink perhaps a little less milk or eat less meat and then in return you can buy better products that this/ that support this whole, this animal welfare.”(15f_<50; 14)

Awareness creation was seen as critical to increase consumers’ readiness to support the new grazing system. Two participants expressed the view that awareness creation for sustainable consumption is a political and educational task and must begin at school because adults lack time for individual research and will thus continue acting as usual. Participants also expressed their belief that there was a positive development in conscious approaches to livestock food purchasing:
“In the future, there will be ever more people who shop more consciously, who also look where the animal comes from and who don’t put [a product] in their shopping baskets in the supermarket simply because it’s cheap.”(10f_50+; 11)

## 4. Discussion

The first objective of this study has been to explore consumers’ perception of the information about the new grazing system. In terms of information communication, the findings suggest that the use of visual elements to evoke positive emotions is a promising strategy for communicating the benefits resulting from the innovative grazing system with virtual fencing. The positive wording and design of the information material was found appealing, which is consistent with previous findings [37,48]. The depiction of the virtual fencing principle and the readable layout of the brochures contributed to most participants’ understanding of the term ‘virtual fencing’, which confirms the suitability of this approach for explaining a complex concept [36]. Interestingly, however, positive pictures of pasture grazing also evoked negative associations with intensive animal farming, which is similar to the findings of Vigors [43]. Negative associations are thus not necessarily caused by information material but can be affected by consumers’ individual values, knowledge about, and involvement in the subjects raised [18,23,42]. This implies that negative associations are unlikely ever to be completely avoided and must, therefore, be addressed in efforts to promote societal acceptance of virtual fencing.

The participants declared that they had low levels of knowledge about agriculture, which is in line with previous research findings [16,19,20]. Such a lack of knowledge can hardly be compensated for by providing bare facts within a declarative knowledge strategy. A possible reason for lack of confidence in their knowledge on the part of consumers is that they have been overloaded with pro-environmental information [22]. Rather than merely presenting facts, therefore, the information can be combined with proposals for action and participation, i.e., by undertaking also a procedural knowledge strategy. This approach was appreciated by the participants of this study, which is consistent with the findings of Carmen et al. [47]. A strategy combining declarative and procedural knowledge, i.e., factual information and suggestions for action, thus seems a more promising approach.

References to norms related to nature and future generations did not resonate well with the participants, although moral norms do reportedly influence preferences for improved animal welfare [40]. Instead, participants enquired after personal benefits, in line with findings by White et al. [39], which often contrasted with public benefits. Furthermore, virtual fencing as a technology was not seen beneficial for individual consumers. Biodiversity conservation supported by livestock systems that use virtual fencing, however, has a potential of being perceived personally valuable. In order to stimulate such perception, moral norms and personal benefits must be mutually linked in communications.

As a second objective, the study assessed the extent of consumers’ understanding of the principle of the new technology of virtual fencing and their associations with this innovation in relation to animal welfare, biodiversity, and pasture-based production. The principle of virtual fencing was generally well understood by the study participants. Consumers reflected on the possible ramifications of the system and evinced ambivalent attitudes with regard to the use of virtual fencing in pasture grazing. Skepticism about the feasibility of virtual fencing and concerns about its impact on animal welfare, on the environment and on human lives were expressed alongside hopes for positive developments in these areas, in line with Eastwood et al. [15]. The connection between virtual fencing and biodiversity conservation was not obvious to the participants, which added to doubts about the usefulness of the technology [39]. The participants’ appraisals of virtual fencing were discussed in terms of the acceptability of specific aspects, such as the strength of electrical stimuli, rather than holistically—a finding also reported in earlier studies on levels of acceptance of modern approaches in dairy farming [16,17]. Concerns about these and other aspects were likely amplified by the perceived insufficiency of the information provided about the technology. As suggested by the findings of Ziamou and Ratneswar [21], however, the inclusion of more technical descriptions might well have had an opposite effect, raising multiple questions and thereby creating further uncertainty about the technology. For example, the use of the word ‘technology’ in agricultural contexts evokes ideas of dramatic interventions in nature which are negatively perceived by consumers [15,16]. Consumers’ ideas about the natural control of grazing animals with regular fences and cowherds often reflected a concept of traditional extensive animal husbandry, which they possibly saw as the only natural system in spite of their concerns about electric fences jeopardizing the safety of wild animals. Such views are common among socially-minded consumers, who are typically more concerned about the impact of a technology on the environment than they are interested in its benefits [16]. Virtual fencing can be seen as a subtle adjustment to traditional methods of pasture grazing that retains natural conditions both for cattle and for wild animals, which is a crucial point for consumer segments less accepting of technological innovation [16]. One of the key communication challenges is to impart this idea to consumers, thereby possibly reducing the polarity in consumers’ assessments of virtual fencing and increasing the likelihood of their accepting the technology.

Finally, the study has provided insights into consumers’ assessments of a grazing system that uses virtual fencing and the products to be derived from it. The findings of the qualitative data collection via TAP are of high value for the development of communication strategies directed to consumers and for marketing of pasture-raised products from novel grazing systems. In regard to consumers’ willingness to support grazing systems implementing virtual fencing through the purchase of pasture-raised products from these systems, this study’s findings suggest that consumers barely perceive any specific benefits of virtual fencing beyond those of pasture-based production in general. Considering this, the results are largely comparable with previous research on ethical consumption. So, pasture-raised products were positively associated with better animal welfare, higher product quality and benefits to human health [26]. The obstacles cited by participants to the purchase of pasture-raised products, such as low availability, perceived associated costs, and low self-efficacy, correspond to earlier findings [34,35]. The willingness of some participants to change their own habits to support environmental protection and animal welfare suggests there is potential for marketing pasture-raised products to relatively broad consumer segments, as earlier found by Weinrich et al. [27]. Future research might usefully focus on the influence of information provision on consumers’ preferences and willingness to pay for products derived from grazing systems implementing virtual fencing and aimed at biodiversity conservation.

This study has several limitations. As is typical of qualitative research, the small sample size excludes the statistical generalizability of the results. Furthermore, the prevalence of participants with a university degree might have biased the results due to the likely effect of academic training on information perception, understanding and assessment. The use of audio- and video-recording during the think aloud protocols, as well as the presence of the interviewer, might have affected the participants’ utterances in terms of social desirability. The verbal protocol method was found useful to gather initial insights into consumers’ perceptions. However, some participants experienced difficulties with concurrent verbalization; in these cases, a structured interview might have been more suitable. Another limitation may have arisen in the way in which the brochures for this study were designed: despite diversifying the information presented using different communication approaches, the topics discussed in the brochures might have had an impact on the categories used in this study. A quantitative investigation based on the findings of this qualitative study may therefore assess the interrelations between consumers’ understanding of the idea of the technology, personal characteristics and attitudes, and consumption patterns.

## 5. Conclusions

Consumers’ interest in products that help foster animal welfare and biodiversity offers a market perspective for farmers who are considering the introduction of pasturing with virtual fencing. In efforts to build consumers’ appreciation for such products, product-specific communications should emphasize the link between personal benefits to consumers, such as higher quality products, biodiversity, landscape conservation, and animal welfare. Emphasis must thus be placed on those aspects that consumers positively associate with pasture grazing, such as better taste, healthiness, and improved animal welfare, while any negative associations with grazing systems that apply virtual fencing should be addressed by making detailed information available on request. The information offered should be concise and easily understandable but soundly grounded in numbers and facts and supported by visual material. Attractive brochures or posters may be a promising tool to gain the attention of consumers at the point of sale. Yet, depending on the place of purchase, additional information should be provided either by well-trained sales personnel or on the internet via easy-to-use interactive sites that can be accessed, for example, using QR codes.

In order to increase consumer awareness of the impact of agriculture on biodiversity, landscape and animal welfare, a dialogue should be established and consumers should be given opportunities to experience directly for themselves the effects of their own actions. To address consumers’ doubts about the impact of their decisions, practitioners and marketers need to demonstrate the direct effects of each individual contribution and convince consumers of the efficacy of achieving change through a step-by-step approach as opposed to adopting an all-or-nothing attitude. For example, given that consumers sympathize with farmers’ hard work but often fail to understand how virtual fencing facilitates its optimization, it would be useful to demonstrate real-life examples as part of participatory communication. One of the aspects to be addressed by policy makers is consumers’ concern about the safety of people who may endanger themselves by approaching grazing cattle in the absence of physical fences. Today, German farmers rely on electric fences as safeguards to avoid potential damage caused by their cattle and related obligation to compensate for it. With the advancement of the novel technology, the approach to liability should be harmonized, and the legal basis should provide farmers with the options to reduce the risk of damage and the risk of liability for the damage caused by cattle.

To conclude, the results of this study demonstrate significant challenges entailed in motivating consumers to support specific practices in animal agriculture through their purchasing actions. Calling for consumer action is pointless without providing options for participation to increase engagement. Any initiative will only be successful if it employs well-targeted communication strategies. In most cases, indeed, it will require considerable effort and political action with substantial governmental support to convert ecosystem services provided by the beef and dairy livestock systems implementing virtual fencing into a private good for which markets can be established.

## Figures and Tables

**Figure 1 animals-10-02267-f001:**
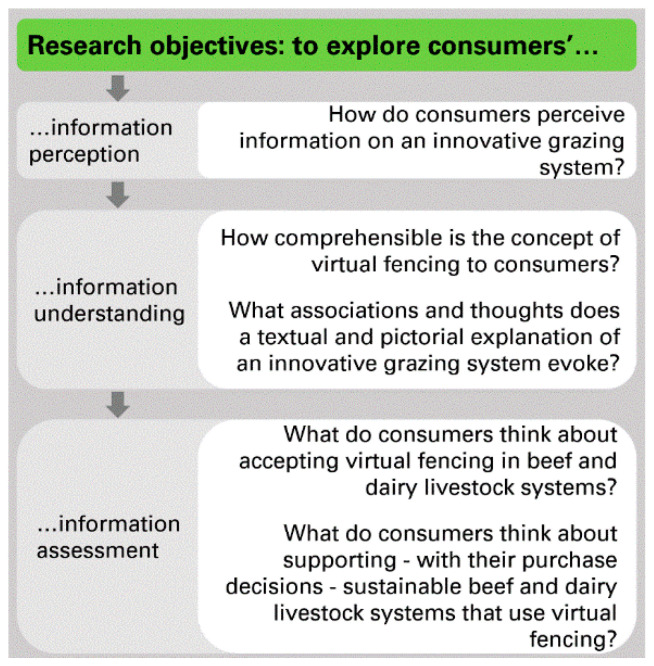
Research objectives and research questions.

**Figure 2 animals-10-02267-f002:**
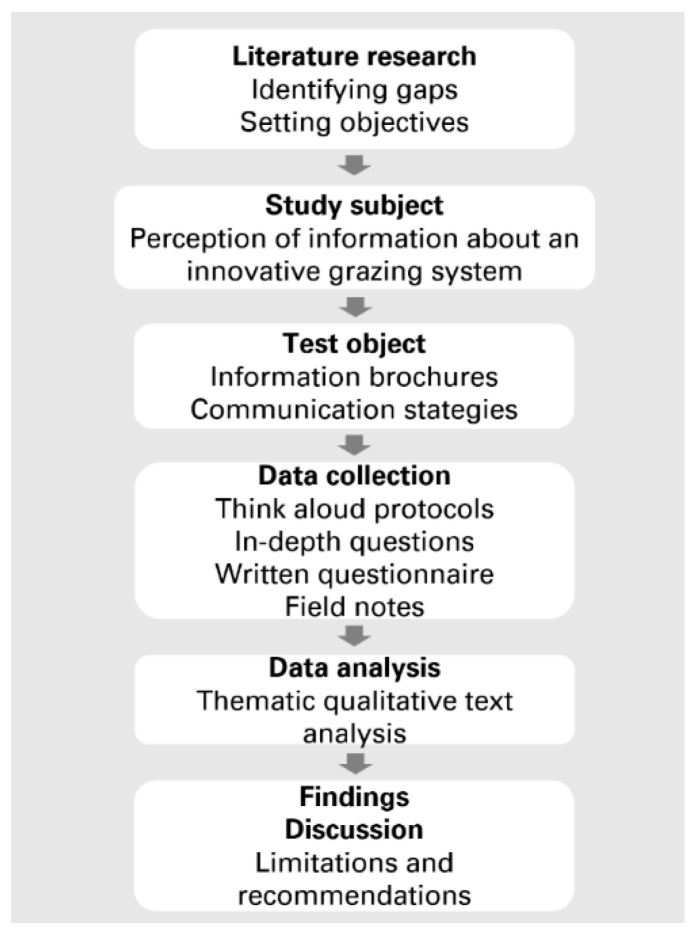
Study design.

**Table 1 animals-10-02267-t001:** Socio-demographic characteristics of the study participants (*n* = 20).

Socio-Demographic Characteristics	Total		Region		
	*n*	%	North-West	East	South
Gender					
Female	12	60	4	4	4
Male	8	40	3	2	3
Age					
18–49 years	10	50	3	3	4
50+ years	10	50	4	3	3
Average age	46.6		46.4	45.2	48.0
Education					
Still in education	0	0			
Incomplete schooling, currently not in education or training	1	5		1	
Primary or secondary education	1	5			1
University entrance qualification or completed vocational training	7	35	2	3	2
University	11	55	5	2	4

**Table 2 animals-10-02267-t002:** Participants’ reasons for negative perceptions of images.

Reason	Example	Sample Excerpt
Image perceived unsuitable to the topic or the layout	Images of a beef steak and a cow alongside each other: the idea of grilling paired with the presentation of the benefits of pasture grazing	“A bit strange, perhaps, to see cattle and then, well, a steak beside them. But that’s how it is.” (15f_<50; 33). “Well, this picture [raw beef steak] scares me. […] I fail to see what that has to do with the subject.” (5m_50+; 18)
Image is confusing	The connection between the depicted objects is not obvious	“I don’t understand why there’s a person with a laptop on the cover. Somehow I find it confusing.” (7f_<50; 13)
Image is found visibly staged, artificial, unrealistic	A lady stroking a cow; a child feeding a cow; a farmer sitting under a tree with a tablet in his hands	“This looks staged to me. That’s not reality anymore.” (17m_50+; 11) “Yes, I think this profession [farmer] here—it’s presented as if it had anything to do with a resting place under a tree. I don’t think this [presentation] has anything to do with real life.” (5m_50+; 32)

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
