# Peer review of "Insights into German Consumers’ Perceptions of Virtual Fencing in Grassland-Based Beef and Dairy Systems: Recommendations for Communication"

_animals, 2020, doi:10.3390/ani10122267_

Round 1

Reviewer 1 Report

The paper is a well-written report of a well-performed qualitative study on a small group of consumers. As such it delivers an interesting insight in the ambiguous feelings of consumers regarding the application of technology to preserve animal wellfare and biodiversity. The results are not very surprising, considering earlier studies in the field. 

I think the study merits publication, although, since the specific subject matter (virtual fencing in relation to consumer perception) is new and relevant.

I only have two smaller remarks:

  • In the introduction, I miss a reference to the work of Boogaard on the different configurations of modernity, tradition and nature in consumer perceptions of (a.o.) dairy farming. The results clearly add to her framework, so a paragraph in the discussion could be added to discuss this.
  • In the sample of respondents (table 1), an overweight can be seen of academically trained people. This may bias the result quite a bit, since they may be trained to judge the information from a specific (more scientific) angle. This is not addressed in the methods or the discussion, as fas as I can see, but it should. 

Author Response

We would like to thank Reviewer 1 for their constructive comments. We highly appreciate the effort that Reviewer 1 put into the feedback on our paper.

The paper is a well-written report of a well-performed qualitative study on a small group of consumers. As such it delivers an interesting insight in the ambiguous feelings of consumers regarding the application of technology to preserve animal wellfare and biodiversity. The results are not very surprising, considering earlier studies in the field. 

I think the study merits publication, although, since the specific subject matter (virtual fencing in relation to consumer perception) is new and relevant.

I only have two smaller remarks:

  • In the introduction, I miss a reference to the work of Boogaard on the different configurations of modernity, tradition and nature in consumer perceptions of (a.o.) dairy farming. The results clearly add to her framework, so a paragraph in the discussion could be added to discuss this.

Response 1: Many thanks for this valuable suggestion. We have now added a reference to the work of Boogaard et al. (2011) in the Introduction (L71–73) and Discussion sections (L528–531 and L 540-544), as below:

L70-73: “Consumers’ perspectives on virtual fencing are as yet unclear, though concerns exist as to its socio-ethical acceptability [15]. This finding is supported, for example, by the results of a study of Dutch citizens’ views on modern farming that indicates a preference for a rather traditional form of ‘natural’ dairy farming [16].”

L528-531: “The participants’ appraisals of virtual fencing were discussed in terms of the acceptability of specific aspects (such as the strength of electrical stimuli, for example) rather than holistically – a finding also reported in earlier studies on levels of acceptance of modern approaches in dairy farming [16,17].”

L540-544: “Such views are common among socially-minded consumers, who are typically more concerned about the impacts of a technology on the environment than they are interested in its benefits [16]. Virtual fencing can be seen as a subtle adjustment to traditional methods of pasture grazing that retains natural conditions both for cattle and for wild animals, which is a crucial point for consumer segments less accepting of technological innovation [16].”

  • In the sample of respondents (table 1), an overweight can be seen of academically trained people. This may bias the result quite a bit, since they may be trained to judge the information from a specific (more scientific) angle. This is not addressed in the methods or the discussion, as fas as I can see, but it should. 

Response 2: Thank you for this comment. We have addressed this point in our “Sample description” (L219-220), and in the Discussion section (L566-568):

L218-219: “Most participants in this study were university graduates.”

L566-568: “Furthermore, the prevalence of participants with a university degree may have biased the results due to the likely effect of academic training on information perception, understanding and assessment.”

Reviewer 2 Report

General comments.

I am not a social scientist so cannot comment on the appropriateness of the approach. My background is animal science, so my comments reflect this. I was hoping the paper would be very interesting as the social acceptance is of great concern to those developing virtual fencing technology, however, I am not sure that this approach as adequately addressed the issue. Interestingly the concerns we have did not rank particularly high with the interviewees: the strength of the electrical stimulus and the failure of the animal to readily understand leading to stress for example? My feeling was that these issues did not arise as the interviewees had a minimal understanding of the technology. The overall conclusion appeared to be that virtual fencing was unlikely to affect consumer sentiment, while I hope this is true, my feeling is that a more rigorous assessment is needed, before we can confirm this conclusion. My view would be that a greater awareness of the technology should have been conveyed to the interviewees before the questionnaire, if you wanted more perceptive responses. As pointed out by the authors, the sample size seemed very small (n = 20) and it would have been interesting to have either increased the sample size or pre-select a demographic more tuned in to the issue. The survey was restricted to Germany, so the title should reflect this. A survey of Australians for example may have provided very different answers. It would be useful if the four trifold posters were included in the publication as an appendix. The nomenclature for quotes left me confused. I assume the first two digits refer to the interviewee (1 thru 20, with a male or female designation), the next digits refer to the age category, but the final digits I have no idea.

Author Response

We would like to thank Reviewer 2 for their constructive critique. We highly appreciate the effort that Reviewer 2 put into the feedback on our paper.

I am not a social scientist so cannot comment on the appropriateness of the approach. My background is animal science, so my comments reflect this. I was hoping the paper would be very interesting as the social acceptance is of great concern to those developing virtual fencing technology, however, I am not sure that this approach has adequately addressed the issue.

Response 1: Thank you for this observation. While we certainly recognize the value of an assessment of social acceptance, such an assessment would require a quantitative study. To provide a basis for such assessment, this qualitative study explored the range of subjects that emerge in connection with the technology in question.

Interestingly the concerns we have did not rank particularly high with the interviewees: the strength of the electrical stimulus and the failure of the animal to readily understand leading to stress for example? My feeling was that these issues did not arise as the interviewees had a minimal understanding of the technology.

Response 2: We agree that a possible lack of understanding of the technology on the part of the interviewees might have affected the spectrum of issues raised by the participants. It was possibly for this reason, for example, that the connection between an animal’s failure to understand a signal and its level of stress was not addressed by the participants, who were rather positive about the animals’ ability to learn and adapt (see L292-294). The participants’ views on electric stimuli – and, their strength in particular – are reported in L377-385.

The overall conclusion appeared to be that virtual fencing was unlikely to affect consumer sentiment, while I hope this is true, my feeling is that a more rigorous assessment is needed, before we can confirm this conclusion. My view would be that a greater awareness of the technology should have been conveyed to the interviewees before the questionnaire, if you wanted more perceptive responses.

Response 3: We completely agree that a more rigorous assessment is needed, as we suggest in L568-570 of the manuscript. We also agree that more extensive information could have contributed to greater participants’ awareness of the technology. However, such density of information might also have led to annoyance and unwillingness to deal with the tested information material. It is one of our conclusions that the amount of information the brochures can provide is insufficient to fully explain the technology, which is why we suggest providing more extensive information on request.

As pointed out by the authors, the sample size seemed very small (n = 20) and it would have been interesting to have either increased the sample size or pre-select a demographic more tuned in to the issue.

Response 4: While we certainly acknowledge that the sample size is small, and perhaps smaller than ideal, it is of an average size for the method applied (Hoppmann 2009). Furthermore, the protocols of the participants interviewed close to the end of data collection did not contribute significantly to the data in that no new data-driven categories emerged during the analysis of their transcripts, thus indicating data saturation. We agree, of course, that a larger sample size would be absolutely necessary for a quantitative study. Regarding the selection of “more tuned in” participants, this would certainly be a requisite for a study of farmers’ perceptions, for example, but the focus of this study was above all on consumers’ perceptions, taking into account that most consumers possess low levels of awareness about animal farming.

The survey was restricted to Germany, so the title should reflect this. A survey of Australians for example may have provided very different answers.

Response 5: This is an important point and we have added the word “German” to the paper’s title (L2).

L2-4: “Insights into German Consumers’ Perceptions of Virtual Fencing in Grassland-Based Beef and Dairy Systems: Recommendations for Communication”

It would be useful if the four trifold posters were included in the publication as an appendix.

Response 6: Thank you for this suggestion. Since the brochures are in German language, we have decided to include them as a supplementary file instead of in an appendix.

The nomenclature for quotes left me confused. I assume the first two digits refer to the interviewee (1 thru 20, with a male or female designation), the next digits refer to the age category, but the final digits I have no idea.

Response 7: The final digits refer to the number of the quoted passage in the transcript of a verbal protocol. We have adjusted the description of the nomenclature of the quotes by adding the word “number” in line L249:

L248-250: “The texts that were read aloud from the brochures are in {curly brackets} and the source (in round brackets) provides the participant’s number, gender, their age group, and the passage number in the transcript.”

Reviewer 3 Report

This is an interesting study.  The paper is well written and easy to read.  I am not very familiar with qualitative analyses.  My expertise in in quantitative statistical analyses.  The conclusions appear to supported by the data and analyses. I was not able to provide any substantive comments or suggestions.  As far as I am concerned the paper is ready for publication.

Author Response

We would like to thank Reviewer 3 for the feedback. We appreciate the time that Reviewer 3 put into reading of our paper.

This is an interesting study.  The paper is well written and easy to read.  I am not very familiar with qualitative analyses.  My expertise in in quantitative statistical analyses.  The conclusions appear to supported by the data and analyses. I was not able to provide any substantive comments or suggestions.  As far as I am concerned the paper is ready for publication.